# Parallel pathways for sound processing and functional connectivity among layer 5 and 6 auditory corticofugal neurons

**Ross S Williamson[1,2†]\*, Daniel B Polley[1,2]**

[1]Eaton-Peabody Laboratories, Massachusetts Eye and Ear Infirmary, Boston, United States; [2]Department of Otolaryngology, Harvard Medical School, Boston, United States

**Abstract** Cortical layers (L) 5 and 6 are populated by intermingled cell-types with distinct inputs and downstream targets. Here, we made optogenetically guided recordings from L5 corticofugal (CF) and L6 corticothalamic (CT) neurons in the auditory cortex of awake mice to discern differences in sensory processing and underlying patterns of functional connectivity. Whereas L5 CF neurons showed broad stimulus selectivity with sluggish response latencies and extended temporal non-linearities, L6 CTs exhibited sparse selectivity and rapid temporal processing. L5 CF spikes lagged behind neighboring units and imposed weak feedforward excitation within the local column. By contrast, L6 CT spikes drove robust and sustained activity, particularly in local fast-spiking interneurons. Our findings underscore a duality among sub-cortical projection neurons, where L5 CF units are canonical broadcast neurons that integrate sensory inputs for transmission to distributed downstream targets, while L6 CT neurons are positioned to regulate thalamocortical response gain and selectivity.
DOI: https://doi.org/10.7554/eLife.42974.001

**\*For correspondence:**
rsw@pitt.edu

**Present address:** †Department of Otolaryngology, University of Pittsburgh School of Medicine, Pittsburgh, United States

**Competing interests:** The authors declare that no competing interests exist.

## Introduction

Corticofugal (CF) neurons broadly fall into two classes: intratelencephalic and sub-cortical (*Harris and Mrsic-Flogel, 2013*; *Harris and Shepherd, 2015*). In the auditory cortex (ACtx), intratelencephalic projection neurons are often found in layer (L) 5a, project locally to L2/3, and distally to ipsi-and contralateral cortex and striatum (*Games and Winer, 1988*; *Winer and Prieto, 2001*; *Yuan et al., 2011*). By contrast, ACtx sub-cortical projection neurons are found in L5b and L6 and target auditory processing centers in the brainstem, midbrain, and thalamus, in addition to telencephalic targets outside of the central auditory pathway, including the striatum and lateral amygdala (*Winer, 2006*; *Bajo and King, 2012*; *Asokan et al., 2018*).

The largest compartment of the auditory CF projection system comes from neurons in layer 5 and 6 of the cortex that innervate the medial geniculate body (MGB) (*Ojima, 1994*; *Prieto and Winer, 1999*; *Winer et al., 2001*). L5 CF neurons deposit giant axon terminal collaterals in the dorsal division of the medial geniculate body (MGBd, (*Ojima, 1994*; *Bajo et al., 1995*; *Rouiller and Welker, 2000*)), en route to additional downstream targets in the tectum, striatum, and amygdala (*Deschênes et al., 1986*; *Bourassa and Deschênes, 1995*; *Bourassa et al., 1995*; *Rockland, 1998*; *Veinante et al., 2000*; *Kita and Kita, 2012*; *Asokan et al., 2018*). L6 CT projections are restricted to the thalamus, with axon collaterals that densely innervate GABAergic neurons in the thalamic reticular nucleus en route to the ventral subdivision of the MGB (MGBv, (*Lund et al., 1979*; *Staiger et al., 1996*; *Zhang and Jones, 2004*; *Llano and Sherman, 2008*; *Guo et al., 2017*; *Cai et al., 2018*; *Chevée et al., 2018*)). Unlike L5 CF neurons, auditory L6 CT axons also collateralize

extensively within the local cortical column, predominantly synapsing onto neurons in layers 5a and 6 (*Prieto and Winer, 1999*; *Guo et al., 2017*).

The distinct projection patterns in L5 CF and L6 CT neurons hint at their potential parallel roles in sensory processing and cortical feedback. L5 CF neurons, with their elaborate dendritic processes and far-ranging axons are regarded as the canonical 'broadcast' neurons of the cortex, pooling inputs from the upper layers and transmitting signals to widespread downstream targets (*Harris and Shepherd, 2015*; *Slater et al., 2019*). L6 CT neurons, by contrast, appear specialized for thalamo-cortical gain control given that their axons terminate in all three nodes of the thalamo-cortico-thalamic loop: the MGB, GABAergic cells in the thalamic reticular nucleus and intrinsically, within the local column (*Olsen et al., 2012*; *Bortone et al., 2014*; *Crandall et al., 2015*; *Crandall et al., 2017*; *Guo et al., 2017*). Differences between L5 CF and L6 CT neurons are not limited to anatomical projections, as additional dichotomies have been described in their cell morphology, intrinsic membrane properties, and synaptic properties (*Diamond et al., 1969*; *Andersen et al., 1980*; *Ojima, 1994*; *Bajo et al., 1995*; *Bartlett et al., 2000*; *Rouiller and Welker, 2000*; *Llano and Sherman, 2009*; *Sherman and Guillery, 2011*; *Sherman, 2016*).

Until recently, it has been difficult to extend these observations to sensory processing in intact, awake animals due to the challenges of recording from identified cell-types in the deeper layers of the cerebral cortex. Here, we leveraged advances in multi-channel extracellular recordings and optogenetics to make targeted recordings from L5 and L6 corticofugal neurons in awake mice. We show that L5 CF neurons utilize dense, non-linear coding of sound features and have little influence on intra-columnar processing whereas L6 CT neurons have sparse selectivity for sound features, and strongly modulate local processing within ACtx. These findings indicate that each class of sub-cortical projection neuron performs distinct operations on incoming sensory signals, which likely impart distinct effects on their downstream targets.

## Results

### Two types of ACtx corticofugal projection neurons

As a first step towards highlighting the differences in L5 CF and L6 CT neurons, we wanted to confirm that the well-established patterns of sub-cortical connectivity in other brain areas and species could be reprised in the mouse auditory system. To visualize L5 CF neurons, we used an intersectional virus strategy to target the divergent axon fields of L5 CF neurons that innervate the tectum (*Asokan et al., 2018*). We first injected canine adenovirus 2 (CAV2-Cre) into the inferior colliculus, which selectively infects local axon terminals to express retrogradely in upstream projection neurons (N = 6 mice, *Figure 1A*: *left*) (*Soudais et al., 2001*; *Schneider et al., 2014*; *Liu et al., 2016*). We then made a second injection of a Cre-dependent virus in the ipsilateral ACtx to label the corticotectal axon fields as well as other potential downstream targets (*Figure 1A*: *left*). As expected, we observed a smattering of smaller neurons in L6b that project to the midbrain, but a majority of cells in L5b with prominent apical dendrites (*Figure 1A*: *left*) (*Schofield, 2009*). In addition to dense terminal labeling in the external and dorsal cortex of the IC (*Figure 1C*: *left*), we also observed axon terminals in the MGBd (*Figure 1B*: *left*), as well as in the striatum and lateral amygdala (*data not shown*, see *Asokan et al., 2018*). These findings confirmed previous reports that corticotectal projection neurons have additional targets in the thalamus and telencephalon, with the caveat that we do not know what fraction of L5 corticotectal cells send collaterals to their different sub-cortical targets (*Moriizumi and Hattori, 1991*; *Bajo et al., 1995*; *Rouiller and Welker, 2000*; *Doucet et al., 2003*; *Chen et al., 2018*).

We used the Ntsr1-Cre transgenic mouse to label L6 CT neurons based on our prior report that 97% of Ntsr1-positive neurons in the ACtx are L6 CT, and 90% of all L6 CT neurons are Ntsr1-positive (*Guo et al., 2017*). We injected a Cre-dependent virus into the right ACtx, allowing a fluorescent reporter to be expressed throughout L6 CT axon fields (*Figure 1A*: *right*). As expected, we observed labelled cell bodies in layer 6a with a band of neuropil labeling in L5a (*Figure 1A*: *right*) and a dense plexus of axon terminal labelling in the MGB, predominantly in the ventral subdivision (*Figure 1D*: *right*). Labelling was not observed in the ipsilateral IC (*Figure 1C*: *right*), indicating that the Ntsr1-Cre expressing cells are separate from other classes of L6 projection neurons that have more distributed local and sub-cortical targets (*Briggs, 2010*; *Sundberg and Granseth, 2018*).

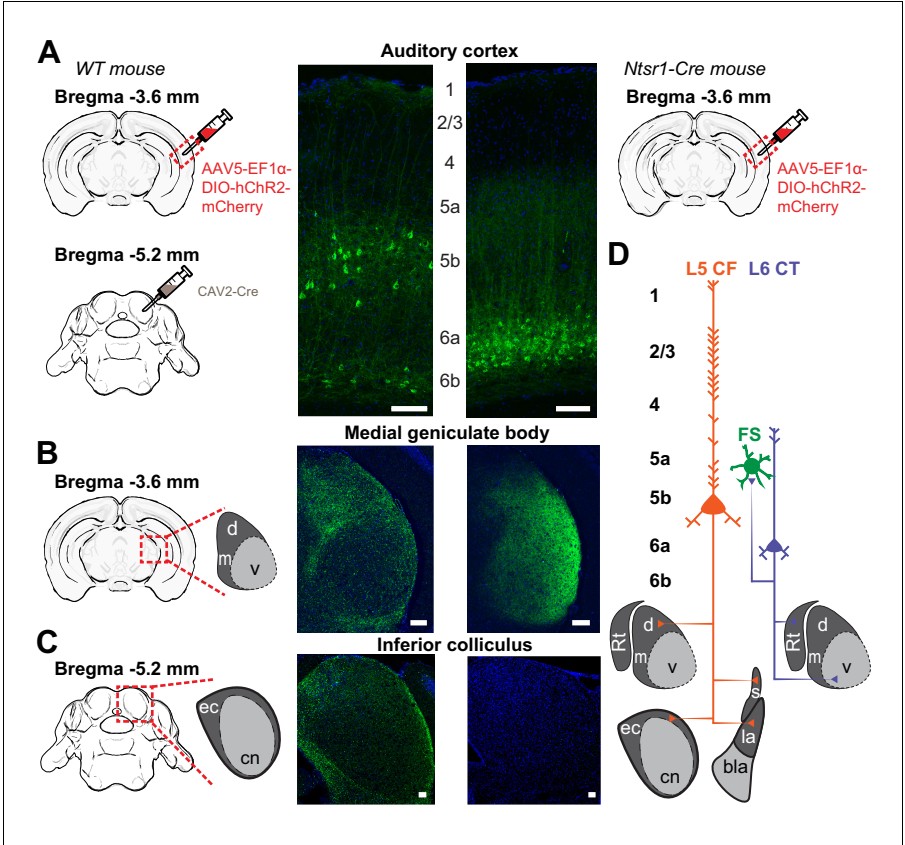

**Figure 1.** Dual corticofugal pathways from L5 and L6. (**A**) Illustration of transgenic and viral strategy used to selectively label L5 neurons that project to the inferior colliculus and MGB (*left*) or L6 neurons that project to the MGB (*right*). Cell body locations and intracortical processes for CAV2-Cre- and Ntsr1-Cre-expressing ACtx neurons. (**B**) 20x confocal sections of the ipsilateral MGB in both L5 CF (*left*) and L6 CT (*right*) mice. The L5 projection shows strong axon labelling in the MGBd, while the L6 projection shows strong axon labelling in the MGBv. (**C**) 20x confocal sections of the ipsilateral IC in both L5 CF (*left*) and L6 CT (*right*). The L5 projection shows strong axon labelling in the external cortex of the IC, while the L6 projection shows none. (**D**) Schematic of dual corticothalamic projection pathways from the ACtx. Abbreviations: d = MGBd, m = MGBm, v = MGBv, Rt = Thalamic Reticular Nucleus, ec = External Cortex, cn = Central Nucleus, s = Striatum, la = Lateral Amygdala, bla = Basolateral Amygdala. Scale bars = 100 μm.
DOI: https://doi.org/10.7554/eLife.42974.002

## Antidromic phototagging of ACtx corticofugal projection neurons

To make targeted recordings from L6 CT neurons, we expressed ChR2 in the ACtx of Ntsr1-Cre mice and implanted an optic fiber such that the tip was positioned dorsal to the MGB (*Figure 2A*). We targeted L5 CF neurons by implanting an optic fiber atop the dorsal cap of the inferior colliculus of WT mice expressing ChR2 in excitatory ACtx neurons (*Figure 2B*). We made extracellular recordings from awake, head-fixed mice (N = 12) using high-density 32-channel silicon 'edge' probes that spanned L5 and 6 (*Figure 2C*). Spikes were sorted into single-unit clusters (n = 1,246) using Kilosort (*Pachitariu et al., 2016*). We used the characteristic pattern of laminar current sinks and sources to assign each recorded neuron to L5 (n = 509) or L6 (n = 625, *Figure 2—figure supplement 1A–B*) (*Kaur et al., 2005*; *Guo et al., 2017*). The peak to trough delay of each unit's waveform was used to classify cells as either regular-spiking (RS) or fast-spiking (FS) (RS: n = 1,028, FS: n = 184, *Figure 2—figure supplement 1C*).

To identify L5 CF and L6 CT projection neurons, we optogenetically activated their axon terminals and documented the temporal patterning of antidromic action potentials (*Figure 2D*). Consistent with previous observations, we hypothesized that antidromically activated spikes with very low trial-to-trial jitter were non-synaptic events reflecting direct photoactivation, whereas high-jitter spikes

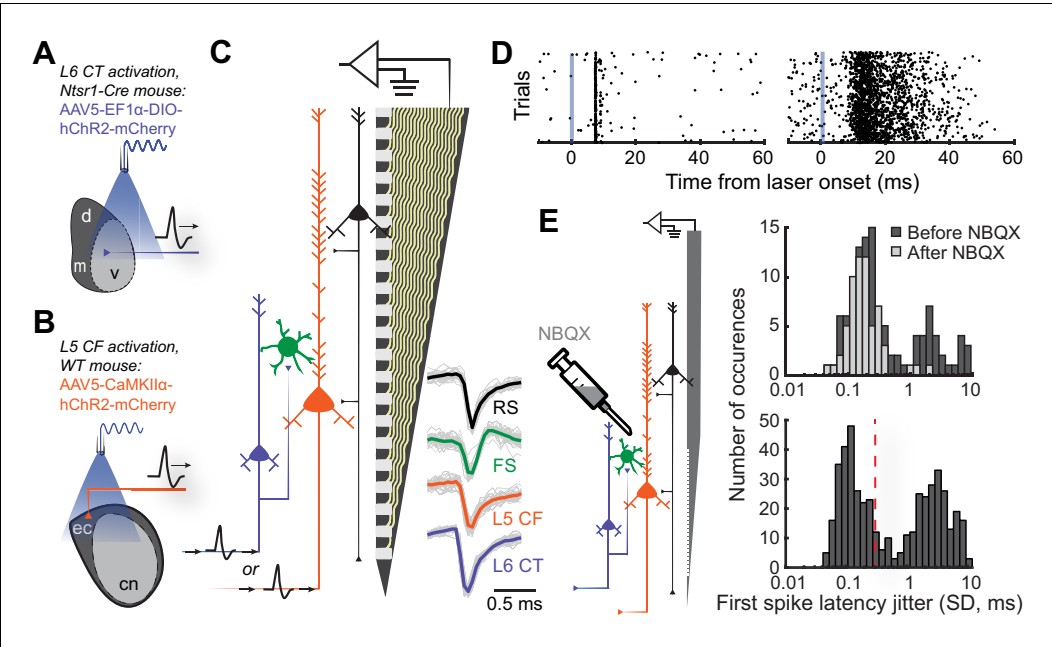

**Figure 2.** Antidromic phototagging of ACtx corticofugal projection neurons. (A) Illustration of the strategy to activate neurons in ACtx L6 that project to the MGB (L6 CT). (B) Illustration of optogenetic approach to isolate L5 ACtx neurons with axons that innervate the IC and MGB (L5 CF). (C) Schematic of the 32-channel silicon edge probe with 20 μm inter-contact spacing used for extracellular recording, alongside four major neuron classes. Example mean spike waveforms (thick lines) are shown with individual waveforms (thin lines). (D) Example unit spike rasters of low- and high-jitter antidromic activity (*left* and *right*, respectively). (E) Illustration of the strategy used to block local synaptic transmission with NBQX (*left*). First-spike latency jitter histograms from anesthetized pharmacology experiments (*top*), and awake experiments showing the 0.3 SD cutoff for identifying directly activated cells (*bottom*).

DOI: https://doi.org/10.7554/eLife.42974.003

The following source data and figure supplements are available for figure 2:

**Source data 1.** Source data for *Figure 2*.
DOI: https://doi.org/10.7554/eLife.42974.006
**Figure supplement 1.** Classifying ACtx neuron identity.
DOI: https://doi.org/10.7554/eLife.42974.004
**Figure supplement 2.** Cell-type separation achieved using measured parameters
DOI: https://doi.org/10.7554/eLife.42974.005

likely arose from intracortical synaptic transmission within the ACtx (*Lima et al., 2009*; *Jennings et al., 2013*; *Li et al., 2015*). To verify this, we blocked glutamatergic transmission within ACtx with local application of NBQX, an AMPA receptor antagonist, in a subset of anesthetized control mice (N = 4, n = 188, *Figure 2E*). NBQX did not affect the low-jitter mode of the distribution but eliminated spikes with variable first-spike latencies, confirming that the low-jitter mode of the distribution reflected direct activation of the recorded projection neuron, whereas higher jitter spikes arose through local polysynaptic activation (*Figure 2E*: *top*). Based on our pharmacology control experiments, we operationally identified recordings of antidromically phototagged L5 CF and L6 CT neurons in our awake recordings as spikes with laser-evoked first-spike latency jitter of 0.3 ms SD or less (L5 CF: n = 132, L6 CT: n = 83, *Figure 2E*: *bottom*). We estimated the laminar locations of the phototagged populations and confirmed that they matched the expected anatomy (*Figure 2—figure supplement 1D*). Few of our phototagged neurons had spike shapes in the FS range (L5 CF: 4.5%, L6 CT: 9.6%), suggesting that these projection neurons are largely separate from the sub-population of parvalbumin-expressing GABAergic projection neurons (*Rock et al., 2018*; *Zurita et al., 2018*). We confirmed that L5 CF units tended to have bursting spike patterns during sensory

stimulation (*Figure 2—figure supplement 1E–F*), as suggested from current injections into L5 neurons recorded in acute slice preparations (*Connors et al., 1982*; *McCormick et al., 1985*; *Agmon and Connors, 1992*; *Joshi et al., 2015*). However, neither spike shape, layer, nor burstiness, could unambiguously identify types of corticofugal neurons from each other or from neighboring L5 and L6 neurons (*Figure 2—figure supplement 2*).

## Sensory characterization of ACtx corticofugal projection neurons

To characterize sensory processing differences in optogenetically identified L5 CF and L6 CT units, we first quantified pure tone frequency response areas (FRAs). Only neurons with well-defined FRAs were included for analysis (L5: n = 174, L6: n = 172, L5 CF: n = 44, L6 CT: n = 27). From each FRA, we computed the tuning bandwidth (20 dB above threshold, *Figure 3A*) and the tone-evoked onset latency (*Figure 3B*). While there were no overall differences in tuning bandwidth between neurons in L5 and 6, L5 CF neurons were more broadly tuned than L6 CT neurons (Wilcoxon Rank Sum test,

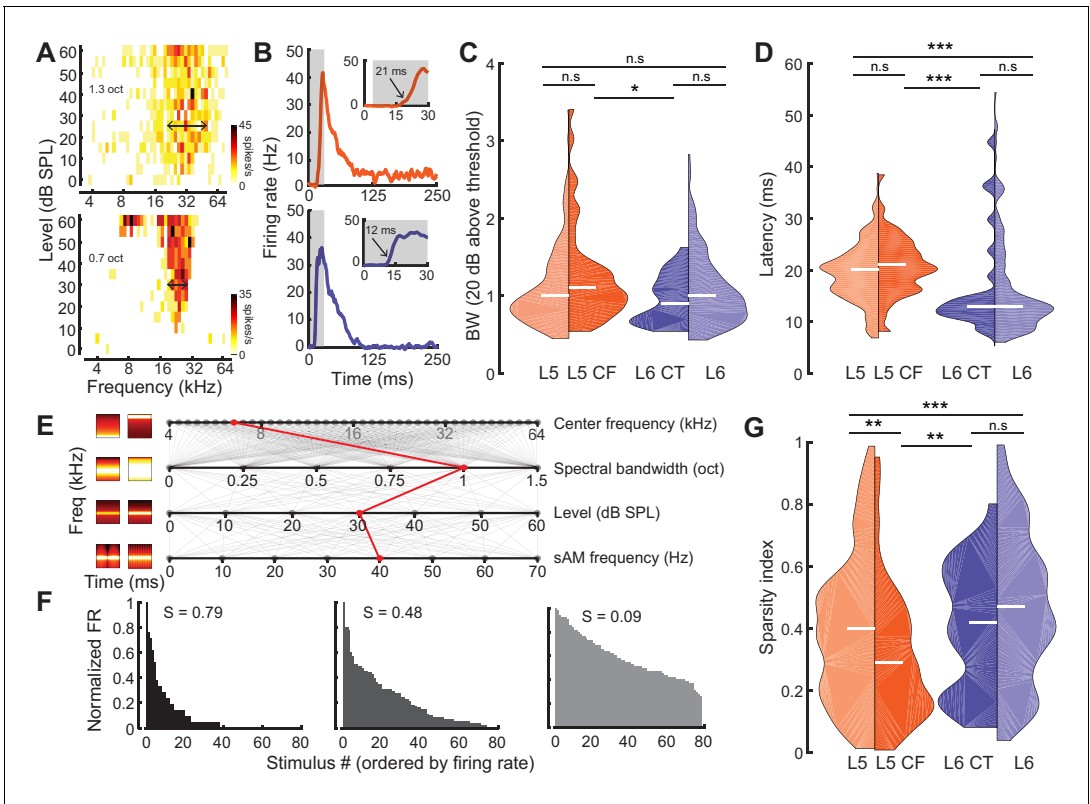

**Figure 3.** Sensory characterization of CF/CT projections. (A) Example FRAs from a broadly-tuned L5 CF neuron (*top*) and a narrowly tuned L6 CT neuron (*bottom*). (B) Example PSTHs from a long-latency L5 CF neuron (*top*) and a short-latency L6 CT neuron (*bottom*). (C) Split-violin plots showing the FRA bandwidth distributions. White horizontal line represents the median. (D) Tone-evoked latency distributions. (E) Spectrotemporally modulated noise burst tokens varied across four acoustic dimensions: Center frequency, spectral bandwidth, level, and sinusoidal amplitude modulation (sAM) frequency, represented here from top to bottom. Frequency x time stimulus spectrograms depict a lower (*left*) and higher (*right*) value for each corresponding acoustic parameter. Red line indicates an example of stimulus values used for a single trial. (F) Example histograms for firing rate across the same 80 randomly selected spectrotemporal noise burst tokens. High sparsity values (S) indicate a narrow distribution with enhanced stimulus selectivity (*left*) while low sparsity values indicate a broad distribution with reduced stimulus selectivity (*right*). (G) Split-violin plots showing the sparsity distributions. Asterisks denote statistically significant differences at the following levels: *p<0.05, **p<0.01, ***p<0.005, as determined by the Wilcoxon Rank Sum test.

DOI: https://doi.org/10.7554/eLife.42974.007

The following source data and figure supplement are available for figure 3:

**Source data 3.** Source data for *Figure 3*.
DOI: https://doi.org/10.7554/eLife.42974.009

**Figure supplement 1.** Correlations between sensory tuning parameters and evoked firing rates.
DOI: https://doi.org/10.7554/eLife.42974.008

p=0.03, *Figure 3C*). Neurons in L5, including L5 CFs, exhibited tone-evoked first-spike latencies that were approximately twice as long as L6 (Wilcoxon Rank Sum test; layer: $p < 2 \times 10^{-9}$, cell-type: $p < 4 \times 10^{-4}$, *Figure 3D*).

As a next step, we generated a set of 80 temporally modulated noise bursts, that varied in center frequency (4–64 kHz, 0.1 octave steps), spectral bandwidth (0–1.5 octaves, 0.1 octave steps), level (0–60 dB SPL, 10 dB SPL steps), and sinusoidal amplitude modulation rate (0–70 Hz, 10 Hz steps) (*Figure 3E*). We then used a standard measure of sparsity to quantify the shape of each cell's response distribution (*Figure 3F*) (*Rolls and Tovee, 1995*; *Vinje and Gallant, 2000*; *Chambers et al., 2014*). This lifetime sparseness index is bounded between 0 and 1, with values close to one reflecting selectivity for a sparse set of stimuli (*Figure 3F*: *left*) and values close to 0 reflecting a broad response distribution (*Figure 3F*: *right*). Responses in L6 were sparser than L5 (Wilcoxon Rank Sum test, $p < 3 \times 10^{-3}$, *Figure 3G*) and complex sound selectivity was sparser in L6 CTs than L5 CFs, in agreement with the differences in pure tone tuning bandwidth (Wilcoxon Rank Sum test, p=0.02, *Figure 3G*). Tuning bandwidth and sparsity were inversely correlated (Pearson's r = −0.3, $p < 2 \times 10^{-8}$, *Figure 3—figure supplement 1A*), but we found no other significant correlations between other tuning parameters (sparsity vs latency: Pearson's r = 0.06, p=0.27, bandwidth vs latency: Pearson's r = 0.08, p=0.07, *Figure 3—figure supplement 1B–C*). Both latency and sparsity were weakly correlated with evoked firing rate (latency vs evoked firing rate: Pearson's r = −0.24, $p < 6 \times 6^{-6}$, sparsity vs evoked firing rate: Pearson's r = −0.21, $p < 3 \times 10^{-6}$, *Figure 3—figure supplement 1D–E*), while bandwidth was not (bandwidth vs evoked firing rate: Pearson's r = 0.01, p=0.84, *Figure 3—figure supplement 1F*).

To determine whether classifying the type of projection neuron improved functional categorization beyond what could be accomplished by reporting the laminar location of the cell body, we fitted a linear support vector machine (SVM) classifier model (*Cortes and Vapnik, 1995*; *Hastie et al., 2008*) to both the CF/CT (*Figure 4A*) and layer (*Figure 4B*) tuning parameters and tested, using cross-validation, how well either the correct cell-type or layer could be predicted. Classification was better than chance (50%) in either case, although classification errors were significantly lower based on projection cell-type than for cortical layer (32% vs 23%, p<0.05, *Figure 4C*). This indicates that variance in sensory response properties can be better predicted by classification of neuron type, rather than just cortical layer.

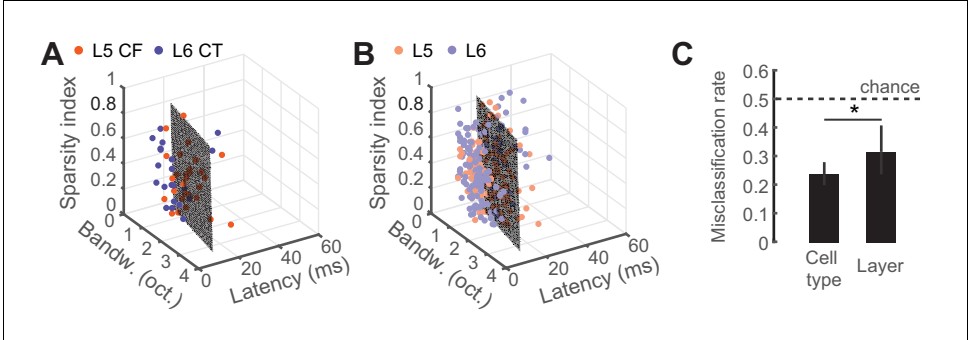

**Figure 4.** Predicting cell-type and layer from neural tuning parameters. (**A**) Scatter plot of tuning parameters for both CF/CT cell-types, with the optimal SVM hyperplane. (**B**) Scatter plot of tuning parameters for L5/L6 cell-types, with the optimal SVM hyperplane. (**C**) Mean (±95% bootstrapped CI) misclassification rates for cross-validated SVM fits to either cell-type or layer. Dotted line is chance, at 0.5. Asterisk denotes a statistically significant difference at p<0.05.

DOI: https://doi.org/10.7554/eLife.42974.010

The following source data is available for figure 4:

**Source data 1.** Source data for *Figure 4*.
DOI: https://doi.org/10.7554/eLife.42974.011

## Modeling the stimulus-response function of ACtx corticofugal projection neurons

Fully characterizing the stimulus-response transformation of a neuron is an intractable problem as it would require knowing the neural responses to all possible stimuli. A common approach is to present a complex stimulus that spans a sizeable subset of the possible stimulus space, and to then relate the stimulus to the resultant neural response using mathematical models. We chose to do this by presenting 20 minutes of a dynamic random chord (DRC) stimulus (*deCharms et al., 1998*; *Linden et al., 2003*). We first computed an estimate of the stimulus-dependent signal power that allowed us to determine which neurons were significantly driven by the DRC (*Sahani and Linden, 2003a*). Only neurons with a significantly non-zero signal power were considered for analysis (L5: n = 372, L6: n = 295, L5 CF: n = 81, L6 CT: n = 51). To mathematically describe the stimulus-response functions for both subtypes of corticofugal projection neuron, we proceeded to fit a multi-linear contextual gain field (CGF) model to the DRC responses (*Ahrens et al., 2008*; *Williamson et al., 2016*). Non-linear contextual effects are prevalent within the auditory system, and this model is capable of capturing some of these known acoustic interactions within the stimulus that can lead to non-linearities within the neural response (*Brosch and Schreiner, 1997*; *Kadia and Wang, 2003*; *Wehr and Zador, 2005*; *Sadagopan and Wang, 2009*; *Phillips et al., 2017*). We quantified predictive accuracy by computing the fraction of the stimulus-related variability that could be captured by the model, and compared this to the accuracy achieved by a linear spectrotemporal receptive field (STRF) model (*Figure 5A*). The CGF model outperformed the linear STRF model on 93% of cells (Wilcoxon Signed Rank test, L5: $p<2\times10^{-52}$, L6: $p<5\times10^{-44}$, L5 CF: $p<2\times10^{-12}$, L6CT: $p<3\times10^{-7}$, *Figure 5B*).

The CGF model combines two receptive field structures. The first is an STRF-like principal receptive field, while the second is the CGF. CGFs consistently featured a suppressive region centered at a zero frequency offset, indicating that preceding sound energy at a similar frequency tended to dampen the impact of a component sound. Regions of enhancement were also present at longer time delays near the preferred frequency, in addition to enhancement in spectral side lobes that overlapped in time with the inputs at the preferred frequency (*Figure 5C-D*) (*Williamson et al., 2016*). Non-linear context effects did not substantially differ between L5 and L6 over frequency or time (mixed effect ANOVA, τ x cell-type interaction, $F_{(1,12)} <0.5$, p=0.95, $\phi$ x cell-type interaction, $F_{(1,24)}<1.3$, p=0.18, *Figure 5D-E*). Among the corticofugal projection neurons however, we noted that the delayed suppressive region (τ) was significantly shorter in L6 CTs, suggesting that the non-linear components of forward suppression operate on faster timescales in L6 CT neurons than L5 CF neurons (mixed effect ANOVA, τ x cell-type interaction, $F_{(1,12)} >2.1$, p=0.02, $\phi$ x cell-type interaction, $F_{(1,24)}<0.3$, p=0.99, *Figure 5D-E*).

## Local connectivity within deep-layer cortical networks

L6 CT neurons have previously been shown to exert a strong feedfoward influence onto the local cortical circuit through interactions with FS interneurons (*Bortone et al., 2014*; *Kim et al., 2014*; *Guo et al., 2017*). To contrast potential differences in the feedforward influence of corticofugal cell-types on intracolumnar processing, we optogenetically activated L5 CF or L6 CT activation and characterized spiking patterns in neighboring RS and FS neurons. Optogenetic activation of L5 CF axons with a 1 ms laser pulse elicited weak, brief polysynaptic activation of deep layer RS and FS units (*Figure 6A*: *left*). By contrast, photoactivating the thalamic terminals of L6 CT units drove powerful, sustained, and distributed activation of local FS and RS units (*Figure 6A*: *right*). We estimated the strength of local synaptic feedforward excitation from L5 CF and L6 CT neurons by quantifying RS and FS firing rates normalized to the peak of direct L5 CF or L6 CT activation. Whereas L5 CF evoked spiking rapidly extinguished in nearby RS and FS neurons, a single pulse of light to L6 CT axon terminals drove a volley of polysynaptic activity lasting approximately 20 ms in local RS and FS cells (Wilcoxon Rank Sum test; L5 CF→FS (n = 96) vs L6 CT→FS (n = 63): $p<4\times10^{-4}$, L5 CF→RS (n = 501) vs L6 CT→RS (n = 296): $p<3\times10^{-25}$, *Figure 6B–C*). Differences in the magnitude and timing of feedforward activation were not reflected in the degree of direct activation of the L5 CF and L6 CT themselves, as the peak firing rates in both cell-types were similar (Wilcoxon Rank Sum test; p=0.64, *Figure 6D*).

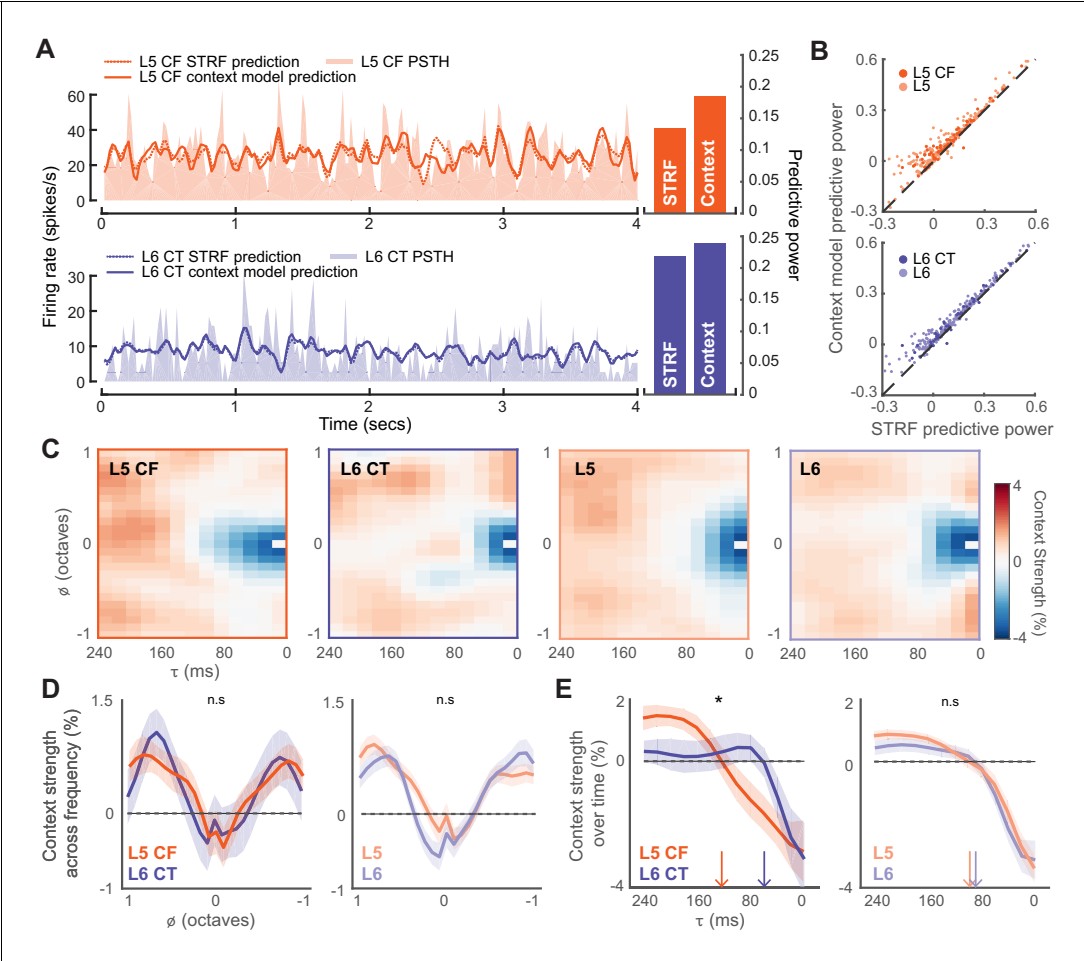

**Figure 5.** Modeling the stimulus-response function of L5 and L6 neurons. (**A**) Snippets of DRC-evoked neural activity with corresponding model predictions for an example L5 CF (*top*) and L6 CT neuron (*bottom*). Predictive powers for the model fits are shown on the right. (**B**) Scatter plots showing the relation between STRF and context model predictive powers for all recorded units. (**C**) Mean contextual gain fields (CGF) for CF/CT and L5/L6 neurons. (**D**) Mean (±1 SEM) of an average across τ for CF/CTs (*left*) and layers (*right*). (**E**) Mean (±1 SEM) of an average across a range of ϕ (between ± 0.25 octaves) for CF/CTs (*left*) and layers (*right*). Vertical arrows depict the extent of the suppressive timescale (the point at which the function crosses zero). Asterisk denotes a statistically significant difference at p<0.05, as determined by a mixed-effect ANOVA.
DOI: https://doi.org/10.7554/eLife.42974.012

The following source data and figure supplement are available for figure 5:

**Source data 1.** Source data for *Figure 5*.
DOI: https://doi.org/10.7554/eLife.42974.014

**Figure supplement 1.** Further quantification of stimulus-evoked variability, model evaluation, and CGF structure.
DOI: https://doi.org/10.7554/eLife.42974.013

We then analyzed the temporal relationship in spiking patterns of both L5 CF and L6 CT neurons and the surrounding RS and FS population. We used a cross-covariance procedure to estimate the strength of temporal interactions during 'steady-state' activation with the DRC stimulus (*Figure 7A–B*) (*Rosenberg et al., 1989*; *Atencio and Schreiner, 2010*). The cross-covariance between RS neurons and both subtypes of sub-cortical projection neurons were relatively weak, although this interaction was stronger for L6 CT neurons (Wilcoxon Rank Sum test; L5 CF↔RS (n = 2353) vs L6 CT↔RS (n = 981): p<5×10$^{-8}$, *Figure 7D–E*). Functional coupling with FS GABAergic interneurons, by contrast, was stronger overall and was particularly pronounced in L6 CT units (Wilcoxon Rank Sum test; L5 CF↔FS (n = 448) vs L6 CT↔FS (n = 209): p<5×10$^{-4}$, *Figure 7D–E*).

As a final step, we calculated the directionality of spike train cross-covariance to test the idea that L5 CFs are integrators that pool inputs from local cells before broadcasting signals out of the cortex,

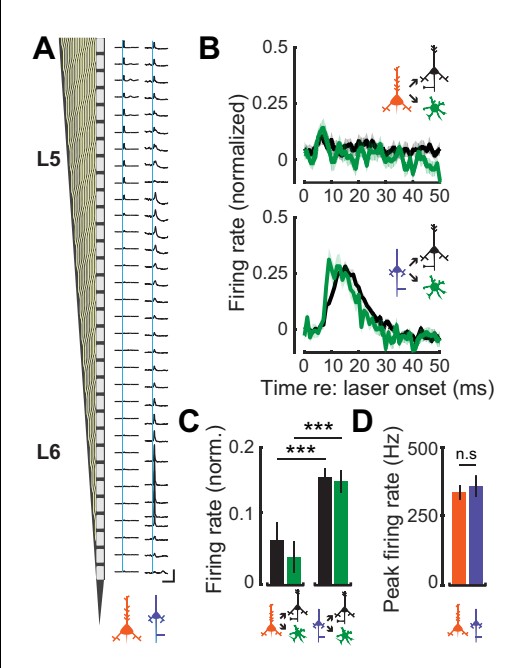

**Figure 6.** Feedforward connectivity within deep-layer cortical networks. (**A**) Schematic of the 32-channel recording probe alongside PSTHs corresponding to each recording site. A 1 ms laser pulse (vertical blue lines) to the axon terminals for L5 CF (left) and L6 CT (right) recruits different patterns of spiking activity in representative deep layer recordings. Vertical scale bar = 200 sp/s; horizontal scale bar = 125 ms. (**B**) Mean (±1 SEM) population normalized PSTHs for both RS and FS groups in response to either L5 CF activation (*top*) or L6 CT activation (*bottom*). (**C**) Mean (±1 SEM) normalized firing rates in local RS and FS units, from the 5–25 ms epoch of the PSTHs shown in *B*. (**D**) Mean (±1 SEM) peak firing rates in L5 CF and L6 CT units. Asterisks denote statistically significant differences at p<0.005, as determined by the Wilcoxon Rank Sum test.

DOI: https://doi.org/10.7554/eLife.42974.015

The following source data is available for figure 6:

**Source data 1.** Source data for *Figure 6*.
DOI: https://doi.org/10.7554/eLife.42974.016

whereas L6 CTs control columnar gain by driving local cell-types. In this case, we hypothesized that L6 CT spikes would temporally lead spikes in local FS and RS cell-types, whereas L5 CF spikes would lag behind local RS and FS spikes. To test this prediction, we first determined which spike train correlations were significant using a confidence bound, before z-scoring the cross-covariance functions and ordering them by the location of their peak (*Figure 7F*). The percentage of lead-preferring interactions were similar across L5 CF and RS/FS pairs, and L6 CT and RS pairs, at approximately 48%. By contrast, 65% of the L6 CT and FS pairs were lead-preferring, underscoring the biased directional interaction between L6 CT neurons and neighboring FS GABAergic neurons. In addition to the temporal peak of the cross-covariance function, we also analyzed the amount of cross-covariance associated with each direction of interaction. We averaged both sides of the z-scored cross-covariance functions before computing a lead-lag asymmetry index by calculating (lead-lag)/(lead+lag) yielding a value whose distance from 0 indicates either lead-preferring (>0) or lag-preferring (<0). We observed that L6 CT spikes tend to lead FS spikes, while the reverse is true for L5 CF spikes (Wilcoxon Rank Sum test; L5 CF↔FS (n = 56) vs L6 CT↔FS (n = 54), $p<8\times10^{-4}$, *Figure 7H*). These results echo previous reports that L6 CT neurons can dynamically regulate the gain of deep layer ACtx networks by driving networks of deep layer FS interneurons (*Bortone et al., 2014*; *Guo et al., 2017*).

## Discussion

Observations made from fixed tissue or in vitro recordings have described marked morphological, intrinsic, and synaptic differences in L5 and L6 projection neurons that suggest the organization of multiple, parallel systems for corticofugal processing (*Diamond et al., 1969*; *Andersen et al., 1980*; *Ojima, 1994*; *Bajo et al., 1995*; *Bartlett et al., 2000*; *Rouiller and Welker, 2000*; *Llano and Sherman, 2009*; *Sherman and Guillery, 2011*; *Briggs et al., 2016*; *Sherman, 2016*). Technical limitations have made it challenging to extend these observations into the realm of sensory processing differences in intact, awake animals while still preserving single spike resolution at a cellular scale. Here, we overcame this technical limitation by leveraging recent advances in multi-channel electrophysiology and optogenetics to make targeted recordings from both L5 CF and L6 CT neurons in awake mice (*Figure 2*).

We showed that L5 and L6 sub-cortical pathways differ in both structure and function. In general, L5 CF neurons exhibited broader spectral tuning and lower lifetime sparseness, indicative of reduced stimulus selectivity and a broader response distribution (*Figure 3*). L5 CF neurons also exhibited a longer timecourse for suppressive stimulus interactions, suggesting an increased sensitivity to local acoustic context (*Figure 5*). Broadly tuned sensory receptive fields could reflect a more extensive

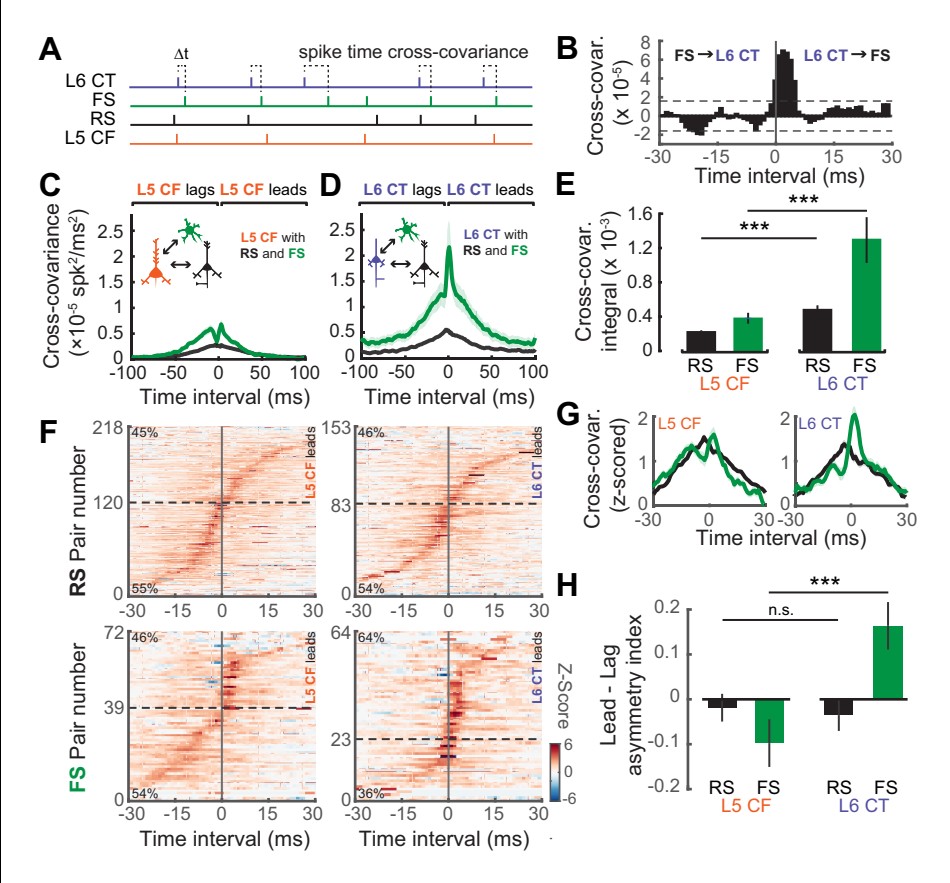

**Figure 7.** Temporal interactions within deep-layer cortical networks. (**A**) Schematic illustrating the four recorded cell-types and the cross-covariance approach. (**B**) An example cross-covariance function highlighting the temporal interaction between a L6 CT↔FS pair. (**C–D**) Mean (±1 SEM) cross-covariance functions of L5 CF (**C**) and L6 CT (**D**) neurons with RS and FS neuron types. (**E**) Mean (±1 SEM) numerical integral of the cross-covariance functions in D-E. (**F**) Significant cross-covariance functions were z-scored and sorted by peak timing. The proportion of pairs where the CF/CT neuron either leads or lags are denoted in the top-left or bottom-left corners, respectively. Dashed horizontal line indicates the separation between leading and lagging pairs. Pairs with a peak at 0 ms were excluded for visualization purposes. Color scale bar denotes the z-scored cross-covariance. (**G**) Marginal distributions of the z-scored cross-covariance functions shown in F. (**H**) Mean (±1 SEM) asymmetry indexes between the positive and negative regions of the cross-covariance functions shown in F, where a value of 0 indicates equivalent regions of leading and lagging cross-covariance, negative values indicate a bias towards CF/CT lagging spike times, and positive values indicate a bias towards CF/CT leading spike times. Asterisks denote statistically significant differences at p<0.005, as determined by the Wilcoxon Rank Sum test.

DOI: https://doi.org/10.7554/eLife.42974.017

The following source data is available for figure 7:

**Source data 1.** Source data for *Figure 7*.

DOI: https://doi.org/10.7554/eLife.42974.018

convergence of intracortical inputs on their elaborate dendritic fields, more broadly tuned excitatory thalamocortical input and narrower inhibitory input received di-synaptically through local fast-spiking interneurons (*Constantinople and Bruno, 2013*; *Hooks et al., 2013*; *Sun et al., 2013*). Such input may lead to these neurons being more sensitive to non-linear effects of acoustic context, consistent with our observations that the nonlinear components of forward suppression operate on longer timescales in L5 CF neurons. Most L5 CF neurons fall into the class of L5 pyramidal neurons referred to as intrinsic bursting (IB), based on their tendency to fire bursts of action potentials in response to depolarizing current injection (*Connors et al., 1982*; *McCormick et al., 1985*; *Llano and Sherman, 2008*; *Joshi et al., 2015*). In vivo whole cell recordings from physiologically identified IB neurons

revealed broader frequency tuning than neighboring cell-types, consistent with our observations of increased frequency tuning bandwidth in bursting L5 CF neurons (*Sun et al., 2013*).

Compared with neighboring corticocortical L6 neurons, L6 CT cells are more densely interconnected with long-range inputs from sensory, retrosplenial, and cingulate cortices (*Vélez-Fort et al., 2014*; *Vélez-Fort et al., 2018*; *DeNardo et al., 2015*), providing a potential source for multisensory response properties (*Morrill and Hasenstaub, 2018*; *Vélez-Fort et al., 2018*). Excitatory and inhibitory neurons in L6 are innervated by thalamocortical axons, suggesting that the sparse tuning of our L6 CT population is likely shaped by strong interactions with the local inhibitory surround, or the temporal relationship between excitation and inhibition (*Zhou et al., 2010*; *Ji et al., 2016*). Indeed, our analyses of functional connectivity showed that the coupling between L6 CT neurons and FS interneurons was stronger than the equivalent L5 CF coupling (*Figure 7*). This is consistent with recent anatomical work in barrel cortex, showing that L6 CT neurons receive more input from local inhibitory neurons than L5 projection neurons (*DeNardo et al., 2015*). Highly selective, sparse sensory tuning has also been described in L6 CT neurons in primary visual cortex, underscoring that the differences described here likely reflect an organizational motif shared across brain areas (*Vélez-Fort et al., 2014*).

Ultimately, these findings shed light on how information contained in two primary output layers of the ACtx can be routed to the rest of the brain. The L6 CT neurons propagate information that is sparse and more selective, but their anatomy places them in a position to modulate information both en route to the thalamus (through the TRN), but also locally through local networks of GABAergic FS interneurons. Indeed, previous work has suggested that these circuits may provide a behaviorally-relevant gating mechanism that can act on thalamic inputs to ACtx (*Yu et al., 2004*; *Zhou et al., 2010*). Such circuitry also allows for rapid modulation of response gain and has led to speculation that L6 CT circuits play a crucial role in dynamically regulating stimulus salience according to internal state variables such as anticipation and attention (*Jaramillo and Zador, 2011*; *Buran et al., 2014*; *Carcea et al., 2017*; *Guo et al., 2017*). Thanks in part to their elaborate dendritic structure, L5 CF neurons are able to integrate input from multiple ACtx layers and broadcast a dense, non-linear signal to multiple sub-cortical auditory stations. CF neurons have been implicated in guiding sub-cortical reorganization of behaviorally relevant sounds (*Gao and Suga, 1998*), enabling experience-dependent learning (*Bajo et al., 2010*), and gating innate defense behaviors (*Xiong et al., 2015*).

Ascending sensory pathways are classically characterized as two streams: a lemniscal system for higher fidelity propagation of detailed stimulus information and a non-lemniscal system that captures contextual sensory influences and internal state variables (*Hu, 2003*; *Lee, 2015*). Here we show that a division of labor between two parallel, complementary streams is maintained in the corticofugal pathway, highlighting a unifying theme in the organization of central sensory pathways.

## Materials and methods

### Mice

All procedures were approved by the Massachusetts Eye and Ear Infirmary Animal Care and Use Committee and followed the guidelines established by the National Institute of Health for the care and use of laboratory animals. This study is based on data from 22 mice (aged 6–8 weeks, both male and female). All mice were maintained under a 12 hr/12 hr periodic light cycle with ad libitum access to food and water. Six C57Bl/6 mice were used for intersectional anatomy experiments (*Figure 1*). Two Ntsr1-Cre mice (B6.FVB(Cg)-Tg(Ntsr1-Cre)-GN220Gsat/Mmcd) and 2 C57Bl/6 mice were used for anaesthetized pharmacology experiments (*Figure 2*). Six Ntsr1-Cre mice and 6 C57Bl/6 mice were used for awake electrophysiology experiments (*Figures 3–7*).

### Surgical procedures
#### Virus-mediated gene delivery
Mice were anesthetized using isoflurane (4%). A surgical plane of anesthesia was maintained throughout the procedure using continuous infusion of isoflurane (1–2% in oxygen). A homoeothermic blanket system (Fine Science Tools) was used to maintain core body temperature at approximately 36.5°C. The surgical area was first shaved and cleaned with iodine and ethanol before being numbed with a subcutaneous injection of lidocaine (5 mg/mL). For viral delivery to the ACtx, an

incision was made to the right side of the scalp to expose the skull around the caudal end of the temporal ridge. The temporalis muscle was then retracted and two burr holes (approximately 0.3 mm each) were made along the temporal ridge, spanning a region of 1.5–2.5 mm rostral to the lambdoid suture. A motorized stereotaxic injection system (Stoelting Co.) was used to inject 0.5 μl of either a non-specific ChR2 viral vector (AAV5-CamKIIa-hChR2[E123T/T159C]-eYFP, UNC Vector Core) or a Cre-dependent ChR2 viral vector (AAV5-EF1a-DIO-hChR2[E123T/T159C]-mCherry, UNC Vector Core) into each burr hole approximately 0.5 mm below the pial surface with an injection rate of 0.05–0.1 μl/min. These viruses were used during optogenetics experiments to target L5 CF or L6 CT neurons, respectively. Following the injection, the surgical area was sutured shut, antibiotic ointment was applied to the wound margin, and an analgesic was administered (Buprenex, 0.05 mg/kg). For intersectional anatomy experiments described in *Figure 1*, CAV2-Cre was injected into the IC via a small craniotomy atop the inferior colliculus (0.5 mm x 0.5 mm, medial-lateral x rostral-caudal, 0.25 mm caudal to the lambdoid suture, 1 mm lateral to midline). Following injections, the craniotomy was filled with antibiotic ointment (Bacitracin) and sealed with UV-cured cement. Neurophysiology experiments began 3–4 weeks following virus injection.

### Implantation of optic fibers

Mice were brought to a surgical plane of anesthesia, using the same protocol for anesthesia and body temperature control described above. The dorsal surface of the skull was exposed, and the periosteum was removed. For awake recordings, the skull was first prepared with 70% ethanol and etchant (C&B Metabond) before attaching a custom titanium head plate (eMachineShop) to the skull overlying bregma with dental cement (C&B Metabond). For L5 CF phototagging, a small craniotomy (0.5 mm x 0.5 mm, medial-lateral x rostral-caudal) was made (0.25 mm caudal to the lambdoid suture, 1 mm lateral to midline), to expose the IC. A ferrule and optic fiber assembly was positioned atop the IC and was cemented to the surrounding skull (C&B Metabond). For mice undergoing L6 CT phototagging, a small craniotomy (0.5 × 0.5 mm, medial-lateral x rostral-caudal) was made under stereotaxic guidance, centered 2.75 mm lateral to midline and 2.75 mm caudal to bregma. A ferrule with a longer optic fiber (~3 mm) was implanted 2.7 mm below the brain surface to activate Ntsr1-Cre positive axons innervating the MGB. Once dry, the cement surrounding the fiber implant was painted black with nail polish to prevent light from escaping. Mice recovered in a warmed chamber and were housed individually.

## Acoustic and optogenetic stimulation

### Acoustic stimulation

Stimuli were generated with a 24-bit digital-to-analog converter (National Instruments model PXI-4461) using custom scripts programmed in MATLAB (MathWorks) and LabVIEW (National Instruments). Stimuli were presented via a free-field electrostatic speaker (Tucker-Davis Technologies) facing the left (contralateral) ear. Free-field stimuli were calibrated before recording using a wide-band ultrasonic acoustic sensor (Knowles Acoustics, model SPM0204UD5).

### Light delivery

Collimated blue light (488 nm) was generated by a diode laser (Omicron, LuxX) and delivered to the brain via an implanted multimode optic fiber coupled to the optical patch cable by a ceramic mating sleeve. Laser power through the optic fiber assembly was calibrated prior to implantation with a photodetector (Thorlabs).

## Neurophysiology

### Awake, head-fixed preparation

Before the first awake recording session, mice were briefly anesthetized with isoflurane (4% induction, 1% maintenance) before using a scalpel to make a small craniotomy (0.5 mm x 0.5 mm, medial-lateral x rostral-caudal) overlying core fields of the ACtx, at the caudal end of the right squamosal suture centered 1.5 mm rostral to the lambdoid suture. A small chamber was built around the craniotomy with UV-cured cement and a thin layer of silicon oil was applied to the surface of the brain. The mouse was then brought into the recording chamber and the head was immobilized by attaching the headplate to a rigid clamp (Altechna). We waited at least 30 min before starting

neurophysiology recordings. All recordings were performed in a single-walled sound-attenuating booth lined with anechoic foam (Acoustic Systems).

At the end of each recording session, the cement chamber was flushed with saline, filled with antibiotic ointment (Bacitracin) and sealed with a cap of UV-cured cement. The chamber was removed and rebuilt under isoflurane anesthesia before each subsequent recording session. Typically, 3–5 recording sessions were performed on each animal over the course of 1 week.

## Data acquisition and spike sorting

At the beginning of each session, a 32-channel, single-shank, silicon probe with 20 µm between contacts (NeuroNexus A32-Edge-5mm-20–177-Z32) was inserted into the ACtx perpendicular to the brain surface using a micromanipulator (Narishige) and a hydraulic microdrive (FHC). Once inserted, the brain was allowed to settle for 10–20 min before recording began. We ensured that recordings were made from the core fields of ACtx (either A1 or AAF) based on the tonotopic gradient, response latencies, and frequency response area shape (*Guo et al., 2012*).

Raw neural signals were digitized at 32-bit, 24.4 kHz (head stage and RZ5 BioAmp Processor; Tucker-Davis Technologies) and stored in binary format for offline analysis. The signal was bandpass filtered at 300–3000 Hz with a second-order Butterworth filter and movement artifacts were minimized through common-mode rejection. To extract local field potentials, the raw signals were notch filtered at 60 Hz and downsampled to 1000 Hz. Signals were then spatially smoothed across channels using a 5-point Hanning window. We computed the second spatial derivative of the local field potential to define the laminar current source density (CSD), which was used to assign each recorded unit to L5 or L6 (*Guo et al., 2017*, *Figure 2—figure supplement 1A–B*).

Spikes were sorted into single-unit clusters using Kilosort (*Pachitariu et al., 2016*). All data files from a given penetration were concatenated and sorted together so that the same unit could be tracked over the course of the experiment (~90 min), and to ensure that a unit drifting across contacts could be accurately assigned to the same cluster. Single-unit isolation was based on the presence of both a refractory period within the interspike interval histogram, and an isolation distance (>10) indicating that single-unit clusters were well separated from the surrounding noise (*Schmitzer-Torbert et al., 2005*; *Harris et al., 2016*).

Once isolated, single units were classified as RS, FS, L5 CF or L6 CT. For the FS and RS classification, the trough to peak interval of the mean spike waveform formed a bimodal distribution, which we used to subdivide neurons with intervals exceeding 0.6 ms as RS, while neurons with intervals shorter than 0.5 ms were FS (*Figure 2—figure supplement 1C*). We used an optogenetic approach to classify units as either L5 CF or L6 CT. A 1 ms laser pulse (with power typically ranging from 10 to 50 mW) was presented between 250 and 1000 times at a rate of 4 Hz, to antidromically activate ACtx neurons. We then fit an analysis window around the laser-evoked spikes in responsive neurons (operationally defined as recordings where the firing rate increased by at least 5 SD from the pre-laser baseline firing rate). We then computed the temporal jitter as the standard deviation of the first spike that occurred within the responsive window (*Figure 2*).

## Anesthetized neuropharmacology experiments

Mice were anesthetized with ketamine (100 mg/kg) and xylazine (10 mg/kg). A homoeothermic blanket system (Fine Science Tools) was used to maintain core body temperature at approximately 36.5°C. A surgical plane of anesthesia was maintained throughout the procedure with supplements of ketamine (50 mg/kg) as needed. An ACtx craniotomy was made as described earlier. In order to block synaptic transmission, we prepared a 1.0 mM NBQX solution by dissolving NBQX (Sigma) into artificial cerebrospinal fluid (ACSF, Harvard Apparatus). After baseline extracellular recordings with a 16-channel silicon probe (NeuroNexus A1×16–100–177–3mm), the NBQX solution was injected approximately 500 µm below the pial surface with an injection rate of 0.05–0.1 µl/min. We then proceeded with optogenetic stimulation experiments after first confirming that sound-evoked local field potential (LFP) responses were qualitatively eliminated.

## Electrophysiological data analyses

### Frequency response areas.

FRAs were delineated using pseudorandomly presented pure tones (50 ms duration, 4 ms raised cosine onset/offset ramps) of variable frequency (4–64 kHz in 0.1 octave increments) and level (0–60 dB SPL in 5 dB SPL increments). Each pure tone was repeated two times and responses to each iteration were averaged. Spikes were collected from the tone-driven portion of the PSTH. First-spike latencies were defined as the time at which the firing rate began to exceed the spontaneous rate by 3 SD. Bandwidths were defined as the FRA width, 20 dB SPL above threshold. A FRA was considered well-defined if it had a d' value of greater than 4. Additional details on FRA boundary determination and d' computation have been described previously (*Guo et al., 2012*). Tone evoked firing rates are summarized in *Supplementary file 1* – supplemental table 3.

### Sparseness

To probe the sparsity of our neuronal population, we generated a bank of 80 random stimuli (*Figure 3E*), in which each stimulus consisted of a noise token that varied across four acoustic dimensions (center frequency (4–64 kHz, 0.1 octave steps), spectral bandwidth (0–1.5 octaves, 0.1 octave steps), level (0–60 dB SPL, 10 dB SPL steps), and sinusoidal amplitude modulation rate (0–70 Hz, 10 Hz steps)). This bank of stimuli was repeated 3–10 times, and the responses averaged.

We used a common measure of sparseness (*Rolls and Tovee, 1995*; *Vinje and Gallant, 2000*; *Chambers et al., 2014*), defined as

$$S_L = \left[ 1 - \frac{\left( \sum_{i=1}^{N} |r_i|/N \right)^2}{\sum_{i=1}^{N} r_i^2/N} \right] / (1 - 1/N) ,$$

where $r_i$ represents firing rates, and $i$ indexes time. This index is defined between zero (less sparse) and one (more sparse), and depends on the shape of the response distribution $p(r)$. Random stimuli evoked firing rates are summarized in *Supplementary file 1*- supplemental table 3.

### Support vector machine classification

To determine how well tuning parameters could be used to correctly decode either cell-type or layer, we used a binary support vector machine classifier with a linear kernel. We fitted the classifier model to a data matrix consisting of N observations of 3 tuning parameters (bandwidth, latency, and sparsity). Each observation was associated with either a positive or negative class (which was used to indicate either cell-type or layer). Ten-fold cross-validation was then used to compute a misclassification rate. The SVM training and cross-validation procedure was carried out in MATLAB using the 'fitcsvm', 'crossval', and 'kfoldLoss' functions. Uncertainty in the misclassification rates was then quantified using a bootstrapping procedure to compute the 95% confidence intervals for both cell-type and layer. A statistically significant difference between groups was established if the classification accuracy of one group fell outside the bootstrapped 95% confidence interval of the other.

### Statistical models of complex sound responses

We presented a 4–64 kHz dynamic random chord (DRC) stimulus (*Linden et al., 2003*). This spectro-temporally rich stimulus is clocked such that every 20 ms a combination of 20 ms cosine-gated tone pulses with randomly chosen frequencies and intensities is generated. Frequencies were discretized into 1/12 octave bins spanning 4–64 kHz, and levels were discretized into 5 dB SPL bins spanning 25–70 dB SPL. The number of tones in each 20 ms chord was random, with an average density of 2 tone pulse per octave. A 1 min DRC was generated, and this was repeated for 20 trials, allowing for 20 min of continuous sound presentation. DRC evoked firing rates are summarized in *Supplementary file 1* – supplemental table 3.

We binned our PSTHs into 20 ms bins (to match the temporal resolution of the DRC) to yield a set of $N$ response vectors $\left\{ r^{(n)} \right\}_{n=1}^{N}$, where $N$ is the number of DRC repeats. We fitted both linear STRF and multi-linear context models to the data. For a stimulus $s$, the linear STRF model is defined as

$$r(i) = c + \sum_{j=1}^{J} \sum_{k=1}^{K} w_{j,k}\, s(i-j+1,k)\,,$$

yielding a prediction of a neural firing rate $r$ at time $i$, with $j$ indexing time and $k$ indexing frequency. This model can be extended to include the effects of acoustic context as

$$r(i) = c + \sum_{j=1}^{J} \sum_{k=1}^{K} w_{j,k}^{t,f}\, s(i-j+1,k)\left(1 + \sum_{m=0}^{M} \sum_{n=-N}^{N} w_{m,n+N}^{\tau,\phi}(s(i-j+1-m,k+n))\right),$$

with $m$ indexing *relative* time and $n$ indexing *relative* frequency. The model consists of a linear component, with a principal receptive field (PRF; analogous to an STRF) denoted by $w^{tf}$, and a contextual gain field (CGF) denoted by $w^{\tau\phi}$, which acts to multiplicatively modulate the stimulus spectrogram prior to spectrotemporal summation by the PRF.

Estimation of the STRF was carried out using the automatic smoothness determination (ASD) algorithm (*Sahani and Linden, 2003b*; *Meyer et al., 2016*). This approach uses regularized linear regression with a spectrotemporal smoothness constraint that is optimized separately for each recording. Estimation of the PRF and CGF in the multi-linear model was carried out using ASD in an iterative alternating least squares (ALS) algorithm (*Ahrens et al., 2008*; *Williamson et al., 2016*).

Following *Sahani and Linden, 2003b*, we defined an estimator for the stimulus-dependent variability within the response, the signal power $P(\mu)$,

$$P(\mu) = \frac{1}{N-1}\left(NP\left(\overline{\boldsymbol{r}^{(n)}}\right) - P\left(\overline{\boldsymbol{r}^{(n)}}\right)\right).$$

An estimate of the stimulus-independent trial-to-trial variability, the noise power, was obtained by subtracting this expression from $P\left(\overline{\boldsymbol{r}^{(n)}}\right)$. The DRC stimulus drove comparable spiking variability in both sub-cortical cell-types (two-sample t-test, p=0.31, *Figure 5—figure supplement 1C*). Noise power was also not different, (two-sample t-test, p=0.73, *Figure 5—figure supplement 1C*) indicating that trial-to-trial variability was similar between groups.

The predictive performance of both models was evaluated using 'predictive power', an explainable variance metric that is normalized by the signal power (*Sahani and Linden, 2003a*). It is therefore a measure of performance that explicitly takes trial-to-trial variability into account, and provides a means to quantify how much stimulus-related variability can be explained by a model. Its expected value for a perfect model is 1, and for a model predicting only the mean firing rate it is 0. Generalization performance of the models was assessed using 10-fold cross-validation.

As described previously, predictive performance on both training and test data depended systematically on the amount of trial-to-trial variability in the recording (*Figure 5—figure supplement 1D–E*) (*Sahani and Linden, 2003a*; *Ahrens et al., 2008*; *Englitz et al., 2010*; *Williamson et al., 2016*; *Meyer et al., 2016*). Following these previous studies, we obtained population-level predictive performance estimates by extrapolating to the 'zero-noise' limit, effectively averaging across the population while discounting the variable impact of noise on each neuron. These extrapolated limits bracket the true average predictive power of the model.

To further investigate the relevance of estimated CGF structure, we quantified the primary modes of variability around the mean using principal components analysis (PCA). In all groups, the scatter around the mean was concentrated in the first two or three principal components (PCs). The first three PCs were able to account for almost 60% of the total variability (*Figure 5—figure supplement 1F*). The structure present within these PCs reflected the consistent features observed in the population CGFs. Namely, the dominant effect in the first PC was to modulate the overall depth of simultaneous/near-simultaneous enhancement, and in the second PC was to modulate the overall depth of delayed suppression (*Figure 5—figure supplement 1G*).

## Functional connectivity

Functional connectivity was assessed using cross-covariance methods (*Rosenberg et al., 1989*). For all pairs of spike trains, we followed *Atencio and Schreiner (2010)* and first computed a cross-correlation function

$$C_{AB}(m) = \sum_{n=0}^{D-m} A(n+m)B(n),$$

where $A$ and $B$ are 1 ms binned spike trains and $D$ is the total duration. This cross-correlation function then leads to an unbiased estimate of the second-order cross-product density, $P_{AB}(m)$ as

$$P_{AB}(m) = \frac{C_{AB}(m)}{D}.$$

Defining the mean intensities of each spike train as $P_A = \frac{N_A}{D}$ and $P_B = \frac{N_B}{D}$,

where $N_A$ and $N_B$ are the total number of spikes in trains $A$ and $B$, respectively, then leads to the cross-covariance function

$$Q_{AB}(m) = P_{AB}(m) - P_A P_B.$$

Upper and lower confidence limits on the cross-covariance functions were set as

$$CL = \pm 3 \left( \frac{P_A P_B}{D} \right)^{\frac{1}{2}}.$$

A cross-covariance function was deemed significant if two consecutive bins satisfied the confidence limit.

## Anatomy

Mice were deeply anesthetized with ketamine and transcardially perfused with 4% paraformaldehyde in 0.01M phosphate buffered saline. The brains were extracted and stored in 4% paraformaldehyde for 12 hr before transferring to cryoprotectant (30% sucrose) for 48 hr. Sections (40 μm) were cut using a cryostat (Leica CM3050S), mounted on glass slides and coverslipped (Vectashield). Fluorescence images were obtained with a confocal microscope (Leica).

## Statistical analyses

All statistical analysis was performed with MATLAB (Mathworks). Data are reported as mean ±SEM unless otherwise stated. Non-parametric statistical tests were used where data samples did not meet the assumptions of parametric statistical tests. In cases where the same data sample was used for multiple comparisons, all p-values remained significant after correction using the Benjamini-Hochberg procedure. Sample sizes have been reported throughout the manuscript and are summarized in *Supplementary file 1* – supplemental tables 1-2.

## Acknowledgements

We thank K Hancock for support with data collection software, W Guo for contributing to electrophysiological data analysis, and L Sheets for guidance with pharmacology. We thank M Pachitariu for both providing and assisting with the use of Kilosort. This work was supported by NIH grants R01 DC017078 (to DBP) and F32 DC01536 (to RSW).

## Additional information

### Funding

| Funder | Grant reference number | Author |
| --- | --- | --- |
| National Institutes of Health | R01 DC017078 | Daniel B Polley |
| National Institutes of Health | F32 DC015376 | Ross S Williamson |

The funders had no role in study design, data collection and interpretation, or the decision to submit the work for publication.

### Author contributions
Ross S Williamson, Conceptualization, Data curation, Formal analysis, Funding acquisition, Validation, Investigation, Visualization, Methodology, Writing—original draft, Project administration, Writing—review and editing; Daniel B Polley, Conceptualization, Supervision, Funding acquisition, Investigation, Methodology, Project administration, Writing—review and editing

### Author ORCIDs
Ross S Williamson (iD) http://orcid.org/0000-0002-5633-7337
Daniel B Polley (iD) http://orcid.org/0000-0002-5120-2409

### Ethics
Animal experimentation: Animal experimentation: All procedures were approved by the Animal Care and Use Committee at the Massachusetts Eye and Ear Infirmary (protocol number 10-03-006) and followed guidelines established by the National Institutes of Health for the care and use of laboratory animals. All surgeries were performed under isoflurane, or ketamine and xylazine, and every effort was made to minimize suffering.

### Decision letter and Author response
Decision letter https://doi.org/10.7554/eLife.42974.022
Author response https://doi.org/10.7554/eLife.42974.023

## Additional files

### Supplementary files
• Supplementary file 1. Supplemental tables detailing sample sizes, parameter values, and firing rates. Supplemental Table S1. Summary table detailing sample sizes and parameter values for all sensory characterization comparisons (*Figures 3–5*). Supplemental Table S2. Summary table detailing sample sizes for all connectivity comparisons (*Figures 6–7*). Supplemental Table S3. Summary table detailing firing rates evoked by all presented stimuli.
DOI: https://doi.org/10.7554/eLife.42974.019

• Transparent reporting form
DOI: https://doi.org/10.7554/eLife.42974.020

### Data availability
All data generated or analyzed during this study are included in the manuscript and supporting files. Source data has been provided for Figures 2-7.

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
