## [Decision Letter]

Thank you for submitting your article "Parallel systems for sound processing and functional connectivity among layer 5 and 6 auditory corticothalamic neurons" for consideration by *eLife*. Your article has been reviewed by three peer reviewers, and the evaluation has been overseen by Andrew King as the Senior and Reviewing Editor. The following individual involved in review of your submission has agreed to reveal his identity: Edward Bartlett (Reviewer #3).

The reviewers have discussed the reviews with one another and the Reviewing Editor has drafted this decision to help you prepare a revised submission.

Summary:

In this manuscript, Williamson and Polley characterize the sensory responses and intracortical influences of the two main corticofugal feedback projections by recording from optogenetically identified neurons originating from layer 5 and from layer 6. They report that layer 5 neurons are more broadly tuned to frequency and other sound parameters, have longer latencies, show longer pre-stimulus contextual effects and have less impact on neighboring cortical neurons than their layer 6 counterparts. They conclude that their data help establish a duality of function of the layer 5 and layer 6 auditory corticothalamic system. This a timely study that uses tour-de-force techniques. The findings presented in this paper represent a significant advance for auditory neuroscience and for understanding laminar information processing and similar dual corticothalamic systems in other sensory cortices.

Essential revisions:

Although the reviewers were enthusiastic in their comments and stated that this is a well conducted study, they did raise several major concerns, which will need to be addressed before a final decision can be made. The principal issue is major point 1.

1) Layer 5 cortico-collicular neurons were phototagged using a retrograde tracer, and it is assumed, based on the authors' previous work (Asokan et al., 2018), that these neurons send branches to the MGB. Although it is acknowledged (subsection “Two types of corticothalamic projection neurons”, first paragraph) that the fraction of corticocollicular neurons that are also corticothalamic is unknown, the fact that the whole manuscript is framed around the layer 5 vs. layer 6 corticothalamic comparison makes this lack of knowledge troublesome. The reviewers found it puzzling that phototagging experiments were performed with a light in the MGB for the layer 6 experiments, but not the layer 5 experiments. Without knowing which of the layer 5 corticocollicular cells are actually corticothalamic cells, this manuscript is essentially comparing a population of corticocollicular cells to another population of corticothalamics: an apples to oranges comparison. This problem could be mitigated if the authors estimate what proportion of corticocollicular cells also send a branch to the thalamus. This could be achieved using anatomical methods or by antidromic activation of layer 5 neurons with light in the MGB to confirm that they are corticothalamic. In the absence of this information, it will be necessary to substantially reframe the conclusions of this study to account for the fact that two groups of neurons with different projection targets (IC and MGB) are being compared.

2) In the assessment of local connectivity (Figure 6), it would be important to know the number of spikes elicited by the antidromic stimulation. Was there a difference between the responsiveness of layer 5 versus layer 6 corticofugal neurons? The authors seem to allude to this in stating that they normalized to the peak of direct activation (subsection “Local connectivity within deep-layer cortical networks”, first paragraph), but it is unclear how this was done or whether the amount of activation was different.

3) In the contextual response fields (Figure 5), it appears that there is more facilitation seen early in the distal spectral "lobes" of layer 5 compared to layer 6 corticofugal neurons (Figure 5C, two left figures), but no difference is seen when the populations are compared quantitatively (Figure 5D, left figure). Did the authors do this comparison over time (but not pooling frequencies as in Figure 5E)?

4) Bursting has been seen in the past in layer 5 corticofugal neurons in slice studies. Assuming that it is feasible to discern bursting using the recording techniques in the current report, was bursting looked for or observed in the layer 5 corticofugal cells reported here?

5) More discussion is needed on the distinct functional contribution of the dual pathways. This should consider (i) potential different inputs to layer 5 and 6 (e.g. their thalamic inputs; Ji et al., 2016); (ii) the functions of layer 5 corticothalamic neurons – a recent study on those corticotectal neurons (Xiong et al., 2015) is relevant to this; (iii) the functions of layer 6 corticothalamic neurons – a previous study in rat A1 suggested they play a role in gating the thalamic inputs to A1.

---

## [Author Response]

Essential revisions:.1) Layer 5 cortico-collicular neurons were phototagged using a retrograde tracer, and it is assumed, based on the authors' previous work (Asokan et al., 2018), that these neurons send branches to the MGB. Although it is acknowledged (subsection “Two types of corticothalamic projection neurons”, first paragraph) that the fraction of corticocollicular neurons that are also corticothalamic is unknown, the fact that the whole manuscript is framed around the layer 5 vs. layer 6 corticothalamic comparison makes this lack of knowledge troublesome. The reviewers found it puzzling that phototagging experiments were performed with a light in the MGB for the layer 6 experiments, but not the layer 5 experiments. Without knowing which of the layer 5 corticocollicular cells are actually corticothalamic cells, this manuscript is essentially comparing a population of corticocollicular cells to another population of corticothalamics: an apples to oranges comparison. This problem could be mitigated if the authors estimate what proportion of corticocollicular cells also send a branch to the thalamus. This could be achieved using anatomical methods or by antidromic activation of layer 5 neurons with light in the MGB to confirm that they are corticothalamic. In the absence of this information, it will be necessary to substantially reframe the conclusions of this study to account for the fact that two groups of neurons with different projection targets (IC and MGB) are being compared.

Yes, this was a logical oversight on our part. To be clear, the anatomy data shown in Figure 1 are from new experiments using different viral labeling than what was published in Asokan et al., 2018. These data replicate the 2018 paper by showing that *at least some* neurons that project to the inferior colliculus also project to the MGB. Logically, some fraction of L5 neurons that were activated by stimulating the collicular terminals also had axon terminals in the MGB (and likely the striatum and other structures as well). As the reviewers point out, we don’t know which of our L5 cells had branched projections and which did not, so we should not have referred to them as L5 corticothalamic. We have remedied this error throughout the revised text and figures by labeling the optogenetically identified L5 neurons as L5 corticofugal (L5 CF). “Corticofugal” is an accurate description that leaves open the possibility for distributed downstream targets without being specific about whether they were corticocollicular and corticothalamic or just corticocollicular.

With regard to the reviewers’ question about why the L5 CT projections were not tagged with fibers in the thalamus – at the time we began these experiments several years ago, we did not know that these corticocollicular neurons would have branched axons in the MGB. We started down this road by attempting dual wavelength phototagging in Ntsr1-Cre mice with a combination of blue and red-shifted opsins to simultaneously identify L5 corticocollicular and L6 corticothalamic neurons. This ended up not working well in our hands so we reverted back to single wavelength phototagging but stuck with our strategy for fiber placement in the MGB of Ntsr1 mice and the IC of WT mice. But now we know that placing the fiber tip in the MGB to tag L5 corticofugal neurons is something that we could focus on in follow-up studies.

2) In the assessment of local connectivity (Figure 6), it would be important to know the number of spikes elicited by the antidromic stimulation. Was there a difference between the responsiveness of layer 5 versus layer 6 corticofugal neurons? The authors seem to allude to this in stating that they normalized to the peak of direct activation (subsection “Local connectivity within deep-layer cortical networks”, first paragraph), but it is unclear how this was done or whether the amount of activation was different.

This is a difficult problem to deal with experimentally, as there are several factors that can introduce variability in the feedforward activation of RS and FS cell types following bulk photoactivation of L5 CF and L6 CT axon fields. For example:

1) Animal to animal differences in fiber placement will impact the number of terminals being activated.

2) Placement of the electrode in ACtx with respect to the location of viral expression.

3) Differences in viral vector tropism, and the number of infected cells.

4) Differences in the total number of L5 CF and L6 CT cells.

5) Differences in the number, strength or cell type-specific targeting of L5 CF and L6 CT neurons onto neighboring neurons in the local column.

We focused our reporting in the Results and Discussion sections on this last possibility. To this end, normalizing the firing rate of downstream RS and FS cells is a good way to mitigate the influences of factors #1-4 because the mean normalized firing rate would reflect consistent differences in the timing and temporal duration of polysynaptic activation from L6 CT and L5 CF cells without being sensitive to absolute spike counts.

The reviewers also make a good suggestion to further address the influence of factors #1-4 by reporting the degree of photoactivated spiking in the L5 CF and L6 CT neurons themselves. We analyzed the peak firing rates of the phototagged L5 CF/L6 CT cells and include these data as a new panel in Figure 6 of the revised manuscript. We found that the peak firing rate was 337 Hz in L5 CF cells and 360 Hz in L6 CT cells, a difference that was not significant. Given the similarity, these new findings underscore that the striking differences in downstream activation of local FS and RS neurons does not simply arise from differences in the degree of activation between L6 CT and L5 CF neurons. We have added a new figure panel (Figure 6D) and added the following text to the manuscript:

“Differences in the magnitude and timing of feedforward activation were not reflected in the degree of direct activation of the L5 CF and L6 CT themselves, as the peak firing rates in both cell types were similar (Wilcoxon Rank Sum test; p=0.64, Figure 6D).”

3) In the contextual response fields (Figure 5), it appears that there is more facilitation seen early in the distal spectral "lobes" of layer 5 compared to layer 6 corticofugal neurons (Figure 5C, two left figures), but no difference is seen when the populations are compared quantitatively (Figure 5D, left figure). Did the authors do this comparison over time (but not pooling frequencies as in Figure 5E)?

The plots in Figure 5D quantify the complete average across /τ (relative time), while the plots in Figure 5E quantify the average across a limited region of /φ (relative frequency). We chose to analyze a limited region of /φ so as to highlight the differences in delayed suppression. The average CGF’s in Figure 5C do show subtle differences in the magnitude and position of the spectral lobes at later time points but the differences are not statistically significant (see Author response image 1 for the L5 CF/L6 CT comparison), so we chose to show the complete average in the figure.

4) Bursting has been seen in the past in layer 5 corticofugal neurons in slice studies. Assuming that it is feasible to discern bursting using the recording techniques in the current report, was bursting looked for or observed in the layer 5 corticofugal cells reported here?

Bursting spike patterns have been described in layer 5 corticofugal cells, though these descriptions have come from direct current injections from intracellular or whole cell recordings, most often in slice preparation. Nevertheless, we attempted to characterize bursting in our in vivo extracellular data, following the analysis methods of Nowak et al. (J Neurophysiol, 2003). We chose to utilize the twenty minutes of dynamic random chord evoked activity, as we felt that this constant stimulation would drive the auditory cortex into a steady state of activity. Following Nowak et al., we then computed log interspike interval histograms for all neurons in our population. Similar to their observations, we found that the ISI distributions exhibited qualitative differences in modality that were indicative of differences in burstiness. We found that skewness provided a reasonably accurate metric to describe the differences we observed. We have expanded Figure 2—figure supplement 1 to show three examples with differing skewness values (Figure 2—figure supplement 1E). The two distributions on the left have skewness values closer to -1. They are both asymmetric and heavy tailed, indicating the presence of “burst modes” of firing. The distribution on the right has a skewness value closer to 0, and is symmetric, indicating an absence of bursting. Figure 2—figure supplement 1F shows the skewness distribution of the population.

We have also added an additional supplementary figure related to Figure 2. We wanted to further make the point that unambiguous cell separation can only be achieved using the phototagging technique that we describe (illustrated in Figure 2—figure supplement 2A). Figure 2—figure supplement 2B-D quantify the separation between distributions of depth, peak-to-trough delay, and skewness to emphasize that these parameters alone do not provide enough information to accurately separate cell class.

Importantly, Figure 2—figure supplement 2D shows the skewness distributions for the different cell classes. Consistent with the numerous intracellular studies, the skewness distribution of the L5 CF population is shifted to more negative values, indicating a greater number of bursting cells.

We have added the following text to the manuscript:

“We confirmed that L5 CF units tended to have bursting spike patterns during sensory stimulation (Figure 2—figure supplement 1E-F), as suggested from current injections into L5 neurons recorded in acute slice preparations (Connors et al., 1982; McCormick et al., 1985; Agmon and Connors, 1992). However, neither spike shape, layer, nor burstiness, could unambiguously identify types of corticofugal neurons from each other or from neighboring L5 and L6 neurons (Figure 2—figure supplement 2).”

5) More discussion is needed on the distinct functional contribution of the dual pathways. This should consider (i) potential different inputs to layer 5 and 6 (e.g. their thalamic inputs; Ji et al., 2016); (ii) the functions of layer 5 corticothalamic neurons – a recent study on those corticotectal neurons (Xiong et al., 2015) is relevant to this; (iii) the functions of layer 6 corticothalamic neurons – a previous study in rat A1 suggested they play a role in gating the thalamic inputs to A1.

We have rewritten paragraphs 2-4 of the Discussion to emphasize these points and cite the work suggested by the reviewers. The second and third paragraphs of our Discussion section details the thalamocortical input to both L5 CF and L6 CT neurons and speculates on how the input relates to the observed tuning properties. We have cited Ji et al., 2016, as well as citing complementary work in the visual system that details the local and long-range inputs to L5 CF neurons. The fourth paragraph of our Discussion section discusses the function of both L5 CF and L6 CT circuits, including citation of the Xiong 2015 paper. We also cite additional papers that suggest L6 CT neurons play a role in gating thalamic inputs to A1.